# Global patterns of the leaf economics spectrum in wetlands

Yingji Pan [1✉], Ellen Cieraad [1], Jean Armstrong[2,3], William Armstrong [2,3], Beverley R. Clarkson [4], Timothy D. Colmer[3], Ole Pedersen [3,5], Eric J. W. Visser [6], Laurentius A. C. J. Voesenek [7] & Peter M. van Bodegom [1]

The leaf economics spectrum (LES) describes consistent correlations among a variety of leaf traits that reflect a gradient from conservative to acquisitive plant strategies. So far, whether the LES holds in wetland plants at a global scale has been unclear. Using data on 365 wetland species from 151 studies, we find that wetland plants in general show a shift within trait space along the same common slope as observed in non-wetland plants, with lower leaf mass per area, higher leaf nitrogen and phosphorus, faster photosynthetic rates, and shorter leaf life span compared to non-wetland plants. We conclude that wetland plants tend to cluster at the acquisitive end of the LES. The presented global quantifications of the LES in wetland plants enhance our understanding of wetland plant strategies in terms of resources acquisition and allocation, and provide a stepping-stone to developing trait-based approaches for wetland ecology.

[1] Institute of Environmental Sciences (CML), Leiden University, Leiden, The Netherlands. [2] Biological Sciences, University of Hull, Hull, UK. [3] School of Agriculture and Environment, The University of Western Australia, Perth, Australia. [4] Manaaki Whenua - Landcare Research, Hamilton, New Zealand. [5] Freshwater Biological Laboratory, University of Copenhagen, Copenhagen, Denmark. [6] Experimental Plant Ecology, Institute for Water and Wetland Research, Radboud University, Nijmegen, The Netherlands. [7] Institute of Environmental Biology, Utrecht University, Utrecht, The Netherlands. ✉email: y.pan@cml.leidenuniv.nl

During the past two decades, trait-based ecology has advanced considerably. The leaf economics spectrum (LES) is an important component thereof. The LES provides convincing evidence of a consistent and continuous relationship among the leaf economics traits, reflecting a gradient of slow (conservative) to fast (acquisitive) strategies in terms of investment and use of nutrients and other resources[1,2]. The LES has been shown to be present across different plant life forms and varied habitat types at a global scale and to a large extent independent of climate[1,3]. Along the LES, species with higher leaf mass per area (LMA) generally have a longer leaf life span (LL), but a lower leaf nitrogen content (leaf N, wt/wt), and lower photosynthetic rates, at least on a mass basis ($A_{mass}$). This conservative strategy usually prevails in less fertile habitats. On the other hand, species with lower LMA, shorter LL, higher leaf N and photosynthetic rate have a faster return on investment of resources, commonly coinciding with nutrient-rich areas. Such trait-trait coordination in LES traits may be caused by underlying physiological and structural trade-offs[4].

Studies on trait-trait relationships, including those on LES, have focused mainly on non-wetland terrestrial plants from a variety of ecosystems, such as forests or grasslands[4,5] or on global analyses[3,6]. However, whether the general LES also exists in global wetland ecosystems still remains unclear. This gap prevails despite the fact that leaf economics traits have been widely measured in wetland plants to study local plant functioning, community structure, growth and competition[7].

A better understanding of trait-based relationships in wetlands is profoundly needed in light of the important ecosystem services provided by wetlands, including their role as the major carbon sink at a global scale[8]. Important ecological processes in wetlands such as methane emission and denitrification are linked to wetland plant functional traits[9,10]. LES traits in wetlands are likely to play a role in these ecosystem processes and services[11,12]. While the wide fertility gradient across different wetland types theoretically provides a natural gradient for the expression of LES from the acquisitive to conservative strategies[13], additional constraints induced by adverse environmental conditions in wetlands compared to non-wetland systems mean that it cannot be taken for granted that LES traits will show similar patterns.

The varied environmental stressors unique to wetland ecosystems constrain plants that inhabit these systems. For example, intermittent/permanent flooding causes altered biogeochemical processes and the production of phytotoxic compounds such as ferrous iron ($Fe^{2+}$) and sulphide ($H_2S$, $HS^-$, $S^{2-}$) in the substrates, as well as a less efficient way of producing ATP in cells experiencing an $O_2$ deficit[14]. In addition, reactive oxygen species (ROS), which can cause cellular macromolecule and membrane damage, accumulate in plant tissues especially upon return to aerobic conditions after flooding[15]. To survive in such an adverse environment, wetland plants have developed a suite of adaptive strategies[15]. Whether the LES also exists in wetlands depends, to a large extent, on whether the prevalent adaptive strategies of plants to environmental stressors are generally costly or cheap[13]. If adaptations are cheap, the LES should be unaffected and similar to non-wetland ecosystems. But if adaptive traits are costly, the LES should be shifted along the same axes (or even shifted in trait space entirely) to compensate this cost[13]. Moreover, leaf mass per area (LMA, one of the LES traits) seems to also be directly involved in flooding tolerance of wetland plants[16], which may also lead to deviations within the LES.

Therefore, our research question is: What is the global leaf economics spectrum in wetlands? And how does it differ from that of non-wetland ecosystems? We hypothesize that wetland plants, in general, follow the LES strategies with fast-return species usually having lower LMA with increased leaf nutrient (N and P) content (wt/wt). This would naturally lead to faster photosynthesis in the day and higher dark respiration rate in the night. Assuming the trade-off between LMA and leaf longevity that exists in non-wetland plants[3] also applies to wetlands, a lower LMA would result in a shorter leaf life span. Despite this general pattern, we also expect that the cost of developing the adaptive traits might affect the trait-trait relationships of leaf economics traits, and consequently shift the overall LES trait pattern in wetlands.

To test these hypotheses, we collected the LES traits measured in 365 wetland species of 184 families from 151 studies of both published and unpublished sources from a global scale. These wetland species are mainly from 10 wetland habitat types (including, as adapted from the Ramsar Convention[17], artificial waterbodies, bogs, estuaries, fens, forested/shrub wetlands, mangrove swamps, marsh, rivers and lakes, temporary brackish/saline non-forested wetlands and temporary non-forested wetlands; see details in Appendix S1 of Pan et al.[18]). These habitat types occupy different positions along the gradients of two dominant drivers: hydrological regime (flooding depth and duration) and fertility (from oligotrophic to eutrophic)[19]. The wetland plant species analyzed in this study represent a full spectrum of plant characteristics and belong to eight life-form categories (emergent, floating-leaved, grass, isoetid, seagrass, sedge, shrub/tree and submerged). To take the effect of submergence on wetland plants into account, we carefully separated traits measured on plants of which only the root-zone or part of the stem was flooded of which tissues emergent above the water table were measured (hereafter called waterlogged wetland plants) vs. traits measured on plant tissues that were submerged (hereafter called submerged wetland plants).

By comparing these trait expressions with an extensive published dataset on non-wetland plants, we examine the trait-trait relationships of LES traits in wetland vs. non-wetland plants. We find that wetland plants, in general, tend to cluster at the acquisitive end of the LES compared to non-wetland plants, with lower leaf mass per area, higher leaf nitrogen and phosphorus, faster photosynthetic rates, and shorter leaf life span. Our global analyses on the LES in wetlands provide a useful perspective on the acquisition and turnover of resources of plants under stressful wetland conditions at a global scale. The results provide a stepping-stone to developing trait-based approaches for wetland ecology. In this way, we can better understand the strategies and functioning of wetland plants from a resource investment/gain perspective[13]. Therefore, studying LES traits in wetlands will not only extend our understanding of global plant strategies on resource acquisition and investment, but also give insight into wetland plant strategies and how these strategies are linked to ecosystem functioning[11,13].

## Results

**Overall bivariate trait-trait relationships.** The overall trait-trait relationships of wetland plants showed similar trends as those among non-wetland plants in terms of the slope directions. Among the significant trait-trait relationships, five out of seven relationships of waterlogged plants had a lower $R^2$ than those of non-wetland plants (such as leaf P vs. leaf N and leaf N vs. LMA), while three out of four relationships for submerged plants had a lower $R^2$ than those of non-wetland plants (Table 1 and Supplementary Table 1). In addition, the confidence interval of $R^2$ of four non-significant trait-trait relationships of both plants of waterlogged and of submerged conditions showed no overlap with the (significant) $R^2$ observed for the corresponding trait-trait relationship of non-wetland plants. In combination, these results indicate weaker trait-trait relationships between wetland plant

**Table 1 Bivariate relationships between leaf traits of the leaf economics spectrum.**

| | log LMA | log $N_{mass}$ | log $P_{mass}$ | log $A_{mass}$ | log $R_{mass}$ | log LL | Plant type |
|---|---|---|---|---|---|---|---|
| log LMA | | –0.61 (–0.68, –0.54) | –1.01 (–1.18, –0.87) | –1.33 (–1.53, –1.17) | –2.24 (–4.45, –1.12) | 1.26 (0.97, 1.65) | Waterlogged |
| | | –1.31 (–1.74, –1.00) | –0.75 (–1.16, –0.48) | –1.98 (–2.54, –1.55) | 0.84 (0.49, 1.44) | – | Submerged |
| | | –0.79 (–0.82, –0.76) | –1.20 (–1.27, –1.13) | –1.30 (–1.38, –1.23) | –1.04 (–1.15, –0.95) | 1.61 (1.50, 1.72) | Non-wetland |
| log $N_{mass}$ | 0.33* (P < 0.001; n = 178) | | 1.60 (1.44, 1.77) | 2.01 (1.67, 2.40) | 2.36 (1.67, 2.40) | –1.89 (–2.96, –1.21) | Waterlogged |
| | 0.23* (P = 0.001; n = 42) | | 1.53 (1.17, 2.00) | 1.30 (0.90, 1.88) | 0.59 (0.34, 1.05) | –1.70 (–43.06, –0.07) | Submerged |
| | 0.57* (P < 0.001; n = 1322) | | 1.51 (1.44, 1.58) | 1.67 (1.57, 1.77) | 1.44 (1.32, 1.57) | –2.10 (–2.25, –1.97) | Non-wetland |
| log $P_{mass}$ | 0.17* (P < 0.001; n = 135) | 0.31* (P < 0.001; n = 264) | | 1.26 (1.01, 1.57) | –1.54 (–3.11, –0.76) | –1.31 (–2.13, –0.80) | Waterlogged |
| | 0.12 (P = 0.123; n = 21) | 0.31* (P < 0.001; n = 41) | | 1.58 (0.62, 4.05) | 0.91 (0.36, 2.30) | –0.91 (–22.96, –0.04) | Submerged |
| | 0.52* (P < 0.001; n = 561) | 0.70* (P < 0.001; n = 555) | | 1.05 (0.91, 1.20) | 0.89 (0.75, 1.06) | –1.00 (–1.14, –0.88) | Non-wetland |
| log $A_{mass}$ | 0.59* (P < 0.001; n = 91) | 0.27* (P < 0.001; n = 90) | 0.12* (P = 0.003; n = 72) | | 1.04 (0.54, 2.00) | –0.71 (–1.16, –0.43) | Waterlogged |
| | 0.56* (P < 0.001; n = 31) | 0.49* (P < 0.001; n = 31) | 0.11 (P = 0.468; n = 7) | | –0.48 (–0.82, –0.28) | – | Submerged |
| | 0.51* (P < 0.001; n = 579) | 0.54* (P < 0.001; n = 537) | 0.19* (P < 0.001; n = 171) | | 0.90 (0.83, 0.98) | –1.35 (–1.43, –1.28) | Non-wetland |
| log $R_{mass}$ | 0.03 (P = 0.626; n = 11) | 0.02 (P = 0.681; n = 11) | 0.13 (P = 0.307; n = 10) | 0.14 (P = 0.255; n = 11) | | 0.53 (0.08, 3.30) | Waterlogged |
| | 0.00 (P = 0.895; n = 16) | 0.27 (P = 0.083; n = 12) | 0.13 (P = 0.423; n = 7) | 0.05 (P = 0.383; n = 16) | | – | Submerged |
| | 0.45* (P < 0.001; n = 228) | 0.58* (P < 0.001; n = 221) | 0.37* (P < 0.001; n = 84) | 0.61* (P < 0.001; n = 220) | | –1.51 (–1.65, –1.38) | Non-wetland |
| log LL | 0.78* (P < 0.001; n = 16) | 0.35 (P = 0.015; n = 14) | 0.34 (P = 0.028; n = 14) | 0.40 (P = 0.021; n = 13) | 0.00 (P = 0.960; n = 4) | | Waterlogged |
| | – | 0.01 (P = 0.949; n = 3) | 0.02 (P = 0.917; n = 3) | – | – | | Submerged |
| | 0.43* (P < 0.001; n = 503) | 0.45*(P < 0.001; n = 489) | 0.27* (P < 0.001; n = 173) | 0.69* (P < 0.001; n = 382) | 0.62* (P < 0.001; n = 187) | | Non-wetland |

The bivariate relationships between leaf traits including leaf life span (LL), leaf dry mass per unit area (LMA), photosynthetic rate ($A_{mass}$), leaf nitrogen (leaf N, wt/wt), leaf phosphorus (leaf P, wt/wt), dark respiration rate ($R_{mass}$) are provided for wetland plants and for comparison given for non-wetland plants. Standardized major axis (SMA) slopes with 95% confidence interval are given in the upper-right section of the table (x variable in column 1, y variable in row 1); coefficients of determination ($R^2$) of SMA and sample sizes are given in the lower-left section of the matrix. The different rows identify statistical properties calculated for waterlogged and submerged wetland plants, and for non-wetland species from the GLOPNET database[3], respectively. The asterisk indicates significant correlation at $P < 0.01$, see 'Methods' for more information.

traits than corresponding relationships among non-wetland plants (Table 1, lower left section), and suggests that wetland plants are less constrained within the LES with larger trait variation. A summary of the results of all standardized major axis (SMA) analyses is given in Table 2.

**Bivariate trait relationships between leaf P, N and LMA.** Leaf P and leaf N were positively correlated, across non-wetland plants[3], waterlogged wetland plants ($R^2 = 0.31$) and submerged wetland plants ($R^2 = 0.31$). The SMA analysis revealed that there was no significant difference in slopes of leaf P-leaf N associations between non-wetland plants and wetland plants ($P = 0.30$ and $P = 0.91$ for waterlogged and submerged wetland plants, respectively). However, the parallel slopes of both waterlogged and submerged wetland plants were elevated compared to non-wetland plants (both $P < 0.001$), which indicates that at a given leaf N, wetland plants tended to have a higher leaf P than non-wetland plants. Moreover, there was a significant shift along the common slope towards higher values in wetland plants (both $P < 0.001$; Fig. 1a). This suggests that the proportional change of leaf P with leaf N of wetland plants was similar to non-wetland plants, while wetland plants generally had higher leaf N and leaf P than non-wetland plants.

Leaf N and LMA were negatively correlated in non-wetland and wetland plants (Table 1). The waterlogged wetland plants had a significantly flatter slope ($P < 0.001$), while submerged wetland plants had a significantly steeper slope ($P < 0.001$). Thus, as LMA decreases, the increase in leaf N was less pronounced in waterlogged wetland plants, while such increase of leaf N was steeper in submerged wetland plants, compared to non-wetland plants (Fig. 1b).

Leaf P and LMA were negatively correlated in both wetland and non-wetland plants with similar slopes ($P = 0.04$ and $P = 0.03$ for waterlogged and submerged wetland plants, respectively, Fig. 1c). Wetland plants had a parallel slope which is shifted towards the upper left corner ($P < 0.001$) compared with non-wetland plants. This indicates that even though leaf P and LMA maintained similar relationships in non-wetland and wetland plants, wetland plants maintained a higher value of leaf P but a lower value of LMA (Fig. 1c).

**Bivariate trait relationships with photosynthetic rate.** The slopes of photosynthetic rate-leaf N in wetland plants were similar to those of non-wetland plants ($P = 0.06$ and $P = 0.18$ for waterlogged and submerged wetland plants, respectively, Fig. 2a). However, waterlogged wetland plants were significantly shifted along a common slope towards a higher photosynthetic rate and leaf N values ($P < 0.001$) and had an elevated parallel slope ($P < 0.001$) compared to non-wetland plants, indicating that at given leaf N, waterlogged wetland plants had a higher photosynthetic rate. This suggests that waterlogged wetland plants had a higher nitrogen use efficiency (photosynthesis per unit investment of leaf N). No significant shift along the common slope nor elevation differences among parallel slopes were detected for submerged wetland plants ($P = 0.61$ and $P = 0.20$, respectively).

There were no significant differences in slopes of photosynthetic rate-leaf P between wetland plants and non-wetland plants ($P = 0.16$ and $P = 0.36$ for waterlogged and submerged wetland plants, respectively Fig. 2b). However, wetland plants of both conditions showed a significant shift along the common slope towards higher photosynthetic rate and leaf P values (both $P < 0.001$). This suggests a similar proportional change between leaf P and photosynthetic rate of both wetland plants and non-wetland plants, while wetland plants had higher values of photosynthetic rate and leaf P than non-wetland plants. No

**Table 2 Comparison of the bivariate relationships in wetland vs. non-wetland plants.**

| | log LMA | | | log N_mass | | | log P_mass | | | log A_mass | | | log R_mass | | |
|---|---|---|---|---|---|---|---|---|---|---|---|---|---|---|---|
| log N_mass | n.a. | n.a. | – | – | – | – | – | – | – | – | – | – | – | – | – |
| | n.a. | n.a. | – | – | – | – | – | – | – | – | – | – | – | – | – |
| log P_mass | <0.001 | <0.001 | <0.001 | 0.274 | 0.274 | 0.137 | – | – | – | – | – | – | – | – | – |
| | <0.001 | <0.001 | <0.001 | 0.896 | 0.896 | 0.357 | – | – | – | – | – | – | – | – | – |
| log A_mass | <0.001 | <0.001 | <0.001 | 0.089 | 0.089 | 0.105 | 0.025 | 0.025 | – | – | – | – | – | – | – |
| | 0.536 | 0.536 | 0.568 | 0.149 | 0.149 | 0.900 | 0.762 | 0.762 | – | – | – | – | – | – | – |
| | n.a. | n.a. | 0.081 | 0.159 | 0.159 | 0.562 | 0.727 | 0.727 | – | – | – | – | – | – | – |
| log R_mass | 0.472 | 0.472 | n.a. | **0.004** | **0.004** | <0.001 | <0.001 | <0.001 | 0.588 | 0.588 | 0.073 | – | – | – | – |
| | <0.001 | <0.001 | 0.725 | 0.498 | 0.498 | 0.046 | <0.001 | <0.001 | 0.030 | 0.030 | 0.821 | – | – | – | – |
| | 0.231 | 0.231 | 0.719 | 0.854 | 0.854 | 0.487 | <0.001 | <0.001 | 0.011 | 0.011 | 0.300 | – | – | – | – |
| log LL | 0.052 | 0.052 | n.a. | n.a. | n.a. | 0.900 | 0.040 | 0.040 | 0.727 | 0.727 | 0.073 | n.d. | 0.784 | 0.217 | – |
| | n.d. | n.d. | 0.725 | 0.498 | 0.498 | 0.562 | 0.134 | 0.134 | 0.821 | 0.821 | **0.001** | 0.099 | 0.298 | 0.695 | – |

The differences in slopes (first column of each trait), shifts along the common slope (second column of each trait) and elevation differences among parallel slopes (third column of each trait) between non-wetland plants vs. submerged wetland plants (first row of each trait) and vs. waterlogged wetland plants (second row of each trait), respectively, were analyzed by SMA. Significant differences are in bold ($P < 0.01$). If slopes are significantly different, this implies differences both in the direction and location of the relationship in trait space[41] (indicated as n.a. = not applicable). Several combinations of leaf life span and other traits at submerged conditions had too few observations to be analysed (indicated as n.d. = not determined).

elevation differences among parallel slopes were detected ($P = 0.10$ and $P = 0.65$ for waterlogged and submerged wetland plants, respectively), suggesting that wetland plants and non-wetland plants have a similar photosynthetic rate per unit leaf P.

The photosynthetic rate-LMA associations were similar between waterlogged wetland plants and non-wetland plants, except for a significant shift ($P < 0.001$) along the common slope towards the corner of lower LMA values but higher photosynthetic rates. This suggests that waterlogged wetland plants generally had lower LMA, but a higher photosynthetic rate. For submerged wetland plants, the photosynthetic rate-LMA slope was significantly steeper than for non-wetland plants ($P < 0.01$). This shows that the decrease of photosynthetic rate with an increase per unit of LMA was stronger in submerged wetland plants, indicating that the effect of changed leaf structure on the photosynthesis was bigger in submerged wetland plants. In other words, the photosynthetic rate of submerged wetland plants was even more reduced by an increase of LMA (Fig. 2c). The significantly different slopes of submerged plants also imply a shift in trait space.

**Bivariate trait relationships with dark respiration rate.** For the relationship between dark respiration rate vs. leaf N, submerged wetland plants showed a significantly flatter slope ($P < 0.01$) than non-wetland plants. This suggests that submerged wetland plants maintained their respiration rate to a lower level as leaf N increases than non-wetland plants (Fig. 2d). Moreover, submerged wetland plants tended to have a lower dark respiration rate at a given leaf P ($P < 0.001$), LMA ($P < 0.001$) or photosynthetic rate ($P < 0.01$) (Fig. 2e–g). In combination, this suggests that for dark respiration, trait relationships are substantially different for submerged plants than for non-wetland plants.

For waterlogged wetland plants, the patterns of dark respiration rate were less apparent. For dark respiration rate vs. leaf N, we found no significant difference in the slopes ($P = 0.15$), nor a shift along the common slope ($P = 0.06$), nor elevation differences among parallel slopes ($P = 0.42$) between the waterlogged wetland plants and non-wetland plants (Fig. 2d). For leaf P, no significant slope ($P = 0.13$) was observed, but waterlogged wetland plants showed a significant shift along the common slope towards higher dark respiration rate and leaf P values ($P < 0.001$) (Fig. 2e). In both cases, the small sample size for dark respiration rate could have reduced the statistical power to detect differences in slope, and particularly in the case of dark respiration rate vs. leaf N the differences in slope was substantial (Supplementary Table 2). Also for dark respiration rate vs. LMA, a $P$-value of 0.03 in combination with the low sample size $n = 11$, suggested the presence of a different slope for wetland vs. non-wetland plants (Fig. 2f). Finally, for photosynthetic rate vs. dark respiration rate, no significant differences were observed for waterlogged vs. non-wetland plants (Fig. 2g).

**Bivariate trait relationships with leaf life span.** How leaf traits co-vary with the leaf life span (LL) in submerged wetland plants remains uncertain, because of the limited number of data points ($n = 3$ for LL-leaf N and LL-leaf P, and the absence of data linking LL-LMA, LL-photosynthetic rate and LL-dark respiration rate). For waterlogged wetland plants, we found a significant lower parallel slopes between LL-leaf N than non-wetland plants ($P < 0.01$, Fig. 3a), suggesting that at a given leaf N, waterlogged wetland plants had a shorter leaf life span. We found no significant differences in the relationships between leaf life span and other traits for waterlogged wetland plants ($P > 0.01$, Fig. 3b–e).

In summary, compared with non-wetland plants, significantly different slopes were detected in the relationship between leaf

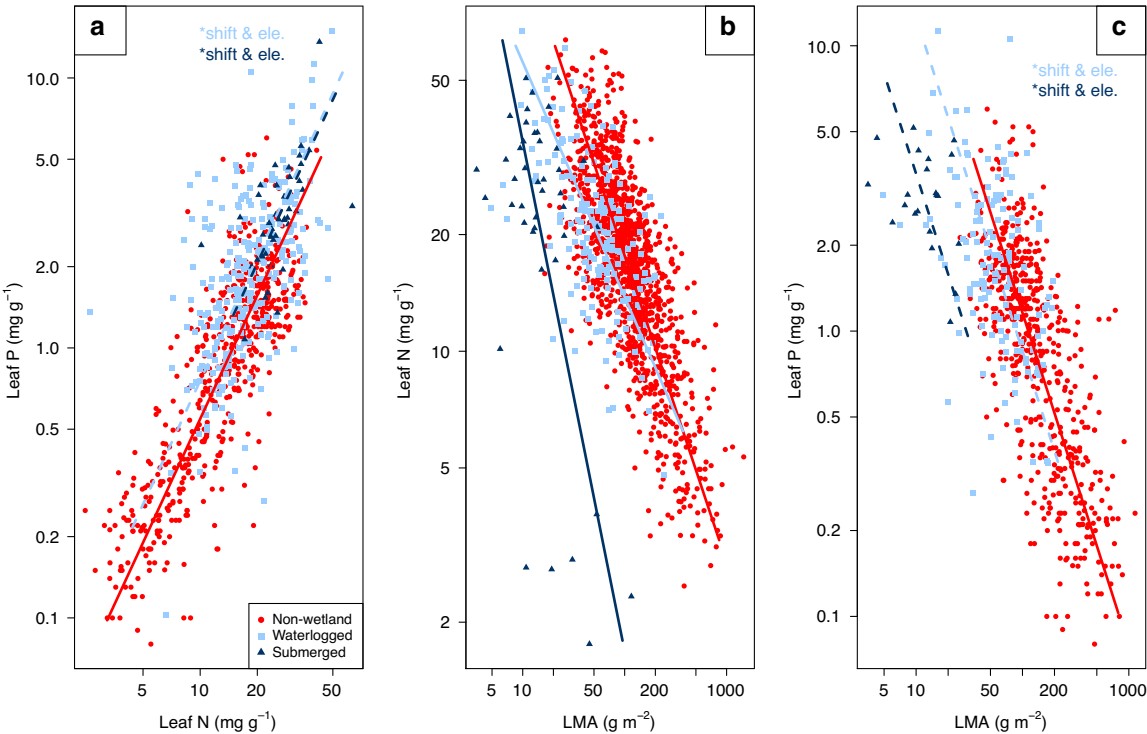

**Fig. 1 The bivariate trait relationships between leaf P, N and LMA. a** Leaf phosphorus (leaf P) vs leaf nitrogen (leaf N). **b** Leaf N vs. leaf dry mass per unit area (LMA). **c** Leaf P vs. LMA. The waterlogged and submerged wetland plants are shown in light blue squares and dark blue triangles, respectively. The non-wetland plant data from GLOPNET[3] are shown in red circles with a solid red line. If the slope for wetland plants differs significantly from that of non-wetland plants, this is indicated by a solid dark or light blue line, for waterlogged and submerged plants, respectively. Dashed lines with the notation of *shift and/or *ele. identify a significant shift along the common slope, and/or significant elevation differences among parallel slopes, respectively. Note that graph axes are $\log_{10}$ scaled.

N-LMA in both waterlogged and submerged wetland plants (Fig. 1b), and between photosynthetic rate-LMA (Fig. 2c) and dark respiration rate-leaf N (Fig. 2d) in submerged wetland plants only. This suggests that submerged wetland plants have even more trait deviations from non-wetland plants than waterlogged wetland plants. In general, wetland plants tended to have a lower LMA with higher leaf N and leaf P contents, and consequently higher photosynthetic rate and shorter leaf life span. For submerged wetland plants, the photosynthetic rate was constrained by an increase in LMA. However, this increase was compensated by a much more gradual increase in dark respiration rate with increasing leaf N, than was evident for non-wetland plants.

## Discussion

We compared leaf economics spectrum (LES) trait associations of wetland and non-wetland plants and found that the LES does exist in wetland plants, but with weaker and often deviating/shifting trait-trait associations relative to the non-wetland LES. The weaker trait-trait associations (as indicated by the lower coefficients of determination ($R^2$) of trait-trait relationships) suggest that alternative strategies exist among wetland plants to deal with the complex and adverse wetland conditions with specific stressors. It may also suggest that besides nutrients and light, other limitations in wetlands also influence the LES and require alternative strategies and consequently the special leaf structure and function of wetland plants. This would cause a higher variation in LES traits. Besides habitat N and P fertility, leaf N can be driven by various factors, including potassium (K), temperature, phytotoxins, or the plants' intrinsic maximal growth rate[7]. Habitat wetness may also drive leaf N through two indirect

mechanisms. On the one hand, denitrification caused by prolonged soil flooding may decrease nitrate availability, thus reducing leaf N[20]. On the other hand, species living in wet habitats usually have a lower LMA, and thus tend to have a higher leaf N[21,22]. The more variable leaf N may further affect the expression of trait-trait associations in wetland plants, such as the leaf N-photosynthetic rate associations[23] and the leaf N-dark respiration rate associations[24].

Our results indicate that the general directions of relationships among LES traits are maintained in wetland plants, which suggests that the principal ecological links behind the trait-trait associations have similarities with those in non-wetland systems[3]. However, our study also reveals differences and these support previous suggestions that wetland plants might possess a unique functional behaviour in photosynthesis-related activities due to their specific adaptation to wetland conditions[25,26]. There are five key aspects in which the LES of wetland plants seems to differ profoundly from the non-wetland LES.

Firstly, in general, wetland plants have a lower LMA, higher leaf N and leaf P content, and a higher photosynthetic rate than non-wetland plants. The waterlogged wetland plants show a shorter leaf life span. Unfortunately, the pattern of submerged wetland plants is uncertain for leaf life span due to a limited number of data points. We conclude that wetland plants comply with a fast-return strategy in resource acquisition among the majority of the LES trait-trait associations[27]. Thus, while nutrient and carbon cycling rates in wetland soils are generally slower compared with non-wetland systems[11], the aboveground carbon and nutrient cycles in wetlands are expected to be faster.

Secondly, a major deviation in LES trait-trait relationships of wetland plants compared to non-wetland plants occurs in the leaf N-LMA relationship (Fig. 1b). The different behaviour of LMA

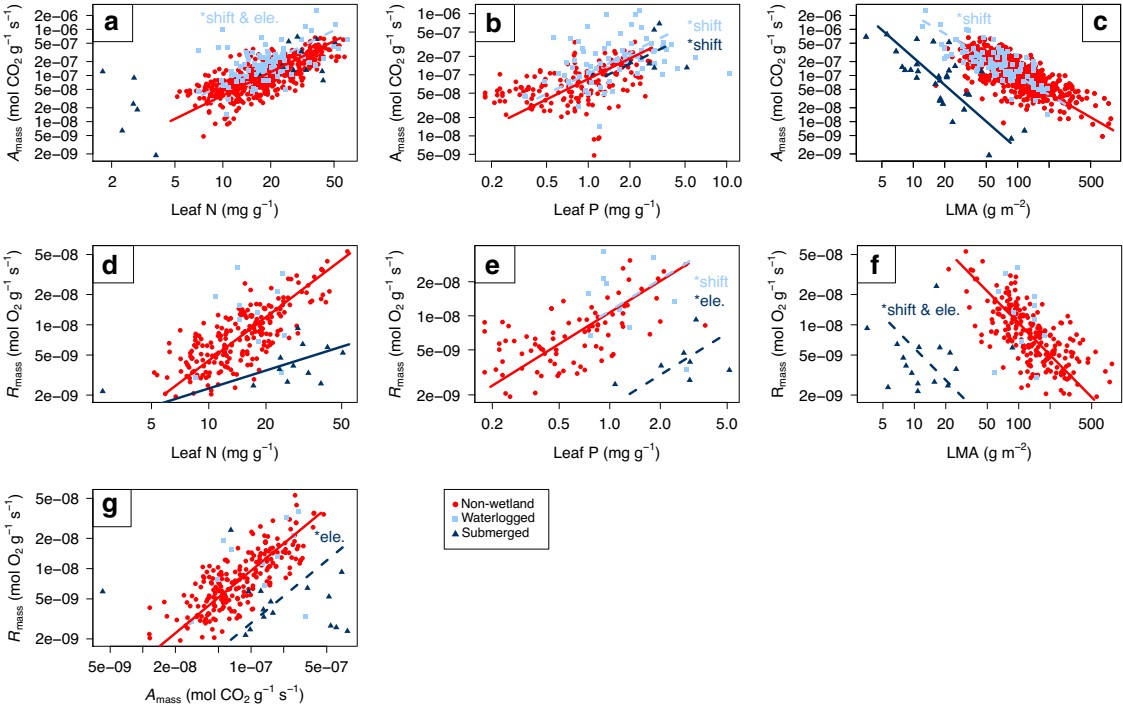

**Fig. 2 The bivariate associations with photosynthetic rate and dark respiration rate. a** Photosynthetic rate ($A_{mass}$) vs. leaf nitrogen (leaf N). **b** $A_{mass}$ vs. leaf phosphorus (leaf P). **c** $A_{mass}$ vs. leaf dry mass per unit area (LMA). **d** Dark respiration rate ($R_{mass}$) vs. leaf N. **e** $R_{mass}$ vs. leaf P. **f** $R_{mass}$ vs. LMA. **g** $A_{mass}$ vs. $R_{mass}$. The waterlogged and submerged wetland plants are shown in light blue squares and dark blue triangles, respectively. The non-wetland plant data from GLOPNET[3] are shown in red circles with a solid red line. If the slope for wetland plants differs significantly from that of non-wetland plants, this is indicated by a solid dark or light blue line, for waterlogged and submerged plants, respectively. Dashed lines with the notation of *shift* and/or *ele.* identify a significant shift along the common slope, and/or significant elevation differences among parallel slopes, respectively. Note that graph axes are $\log_{10}$ scaled.

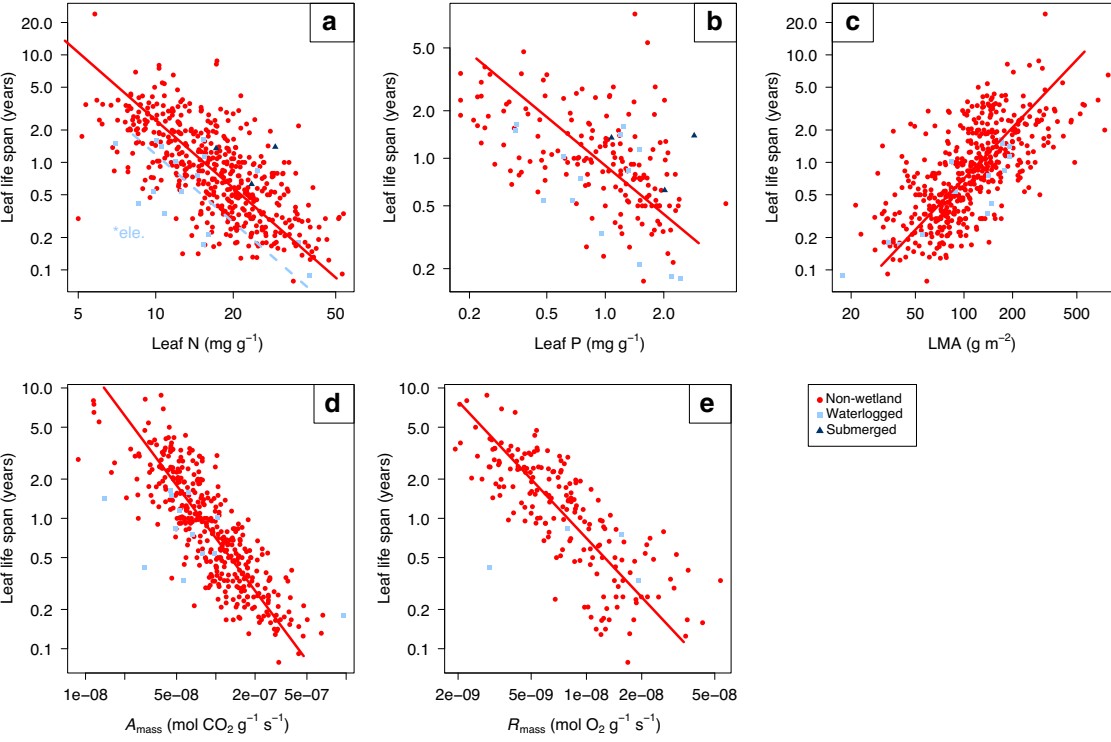

**Fig. 3 The bivariate relationships with leaf life span. a** Leaf life span vs. leaf nitrogen (leaf N). **b** Leaf life span vs. leaf phosphorus (leaf P). **c** Leaf life span vs. leaf dry mass per unit area (LMA). **d** Leaf life span vs. photosynthetic rate ($A_{mass}$). **e** Leaf life span vs. dark respiration rate ($R_{mass}$). The waterlogged and submerged wetland plants are shown in light blue squares and dark blue triangles, respectively. The non-wetland plant data from GLOPNET[3] are shown in red circles with a solid red line. The dashed light blue line with the notation of *ele.* identifies significant elevation differences among parallel slopes respectively. Note that graph axes are $\log_{10}$ scaled and the absence of leaf life span data coupled to LMA, $A_{mass}$, or $R_{mass}$ for submerged plants.

highlights the different functional role of LMA in wetland plants[16,28]. This complies with experimental studies that have found some low-LMA leaves of hydrophytic wetland plants to be functionally highly acquisitive[21,22]. However, in addition to further stimulating the acquisition of nutrients, we also expect that a lower LMA is essential to deal with the lower $CO_2$ and $O_2$ availabilities to the leaves in (partially) submerged conditions[29]. Therefore, besides its leaf economics aspect, LMA should be considered also as a key wetland trait.

Thirdly, a lower LMA may have important implications for the functioning of the remainder of the LES in wetland conditions. In non-wetland low nutrient conditions, plants tend to conserve their nutrients by increasing their LMA to protect the leaves against herbivory and other damages[30]. Our results suggest that such protection of leaves is not feasible in wetlands. In addition, the higher leaf N in wetland plants may also cause an increased risk of herbivory[31]. Together, these processes partially explain the higher herbivory rates in wetland ecosystems compared to non-wetland terrestrial ecosystems[31]. One way to compensate for the higher losses is to become more acquisitive. Such a strategy is supported by the shift in LES traits along the common slope, but may also relate to the elevated leaf P at a given LMA. The results on the leaf N to leaf P relationships suggest that leaf P is even more elevated in wetland plants than leaf N (Fig. 1a). Through these changes in leaf nutrient economics, wetland plant species may raise their photosynthetic capacity in order to create faster growth dynamics (and concomitant higher turnover).

Fourthly, wetland plant species seem to go even further in stimulating photosynthetic capacity. The photosynthetic rate of waterlogged plants was elevated at a given leaf N compared to the photosynthesis-leaf N relationships in non-wetland plants. Leaf N (and leaf P) expresses the combination of photosynthesis-related active nutrients and those nutrients used for storage and protection[32]. If wetland plants indeed invest less energy in the protection of their leaves, the fraction of nutrients involved in photosynthesis increases[4], which in turn would explain the elevated photosynthetic rate of waterlogged plants that we observed. The lower LMA itself may also influence the leaf N-photosynthetic rate relationships, thus increasing the leaf N efficiency of photosynthesis[23]. Finally, some submerged aquatic plants are able to enhance their photosynthesis with special leaf structure, such as thin cuticles and oriented chloroplasts towards the epidermis[22,25].

Lastly, leaves of submerged wetland plants have a lower dark respiration rate (mass basis) than expected from a comparison with the non-wetland LES. Oxygen can decline to hypoxic levels during submergence, and especially in shallow water bodies during the night[33]. Low oxygen can restrict aerobic respiration, both in roots[34] and in leaves[35]. The relatively low dark respiration rate in leaves of wetland plants may be due to a lower investment of resources in leaf construction and maintenance, and related reductions of energy requirements and respiration during the night[24]. The lower respiratory demand allows to more readily face hypoxia when leaves become submerged. In addition, leaves with porous tissues will enhance the oxygen status of the innermost cells. Note that, although the adaptive formation of aerenchyma will significantly decrease the cell oxygen consumption on a tissue volume basis[36], the data analysed here are measurements expressed on a tissue mass basis. Hence, aerenchyma formation per se does not explain the patterns found in this study.

Some of these mechanisms may be further amplified at submerged conditions, where we additionally observed that the altered leaf structure may also affect the photosynthetic rate through a deviating $A_{mass}$-LMA relationship (Fig. 2c), and through influencing the respiration rate by deviating $R_{mass}$-leaf N associations (Fig. 2d). We found a significant reduction of the photosynthetic rate at a given LMA in submerged wetland plants. The additional limitation to photosynthesis of submerged wetland plants can be due to the much lower light availability with water depth and turbid water[29]. However, the unique adaptive traits evolved in wetland plants such as leaf gas films and aerenchyma tissues should enhance the gas exchange/flux in plant tissues[15,35], and therefore partially compensate the costs posed by the adverse wetland conditions. This may explain the observed pattern that the photosynthetic rate at a given leaf N and leaf P-value was not affected (Fig. 2a, b).

All of the described significant changes in the slope of trait-trait relationship, in the position along the slope or due to shifted parallel slopes were detected based on a rather conservative P-value threshold ($P < 0.01$) in this study. This threshold was chosen to help ensure that the most ecologically relevant relationships were detected in relatively large datasets (e.g. a relationship with an $R^2$ of only 0.05 is already significant at $P = 0.05$ at a sample size of $n = 77$). However, for those relationships with smaller sample sizes (in particular in relation to dark respiration rate and leaf life span), this approach may have resulted in overly conservative interpretation. This indicates that deviations in the LES of wetland plants may include even more trait-trait relationships than identified here. Future research could serve to quantify whether these patterns are robust in the face of more data.

Altogether, our analysis suggests that the direct link between photosynthetic rate and dark respiration rate, as evidenced from non-wetland plants to complement N-rich enzymatic and other metabolic components that lead to a higher respiration cost when maintaining a high photosynthetic rate[24,37], also exists in wetland plants. However, such a relationship is expressed differently in wetland plant species compared with non-wetland plants. The results from our analysis show that submerged wetland plants are capable of having lower dark respiration rate at a given photosynthetic rate than non-wetland plants.

When upscaling the findings to wetland ecosystem functioning, we ascribe the generally high productivity in wetland ecosystems globally to the adaptation of wetland plants by having generally fast-return strategies and a higher payback rate. In this way, the adverse wetland conditions may have very limited impact on the wetland plant functioning in terms of resource accumulation. The assumed trade-offs between the cost of adaptation to wetlands and plant function from the leaf economics spectrum perspective are therefore not profound in general[13]. In addition, there are some environmental stressors that rarely happen in wetlands. For example, drought stress, which is a common problem in terrestrial ecosystems, is less constraining in most wetlands, and might move LES traits of wetland plants to the optimum end with lower LMA with higher leaf nutrient content[16,38]. The combination of the high productivity in wetlands and the retarded biochemical cycling rate in the anoxic environments of the substrates together make wetlands the largest contributor to the terrestrial biological carbon pool[8].

## Methods
**Data compilation**. We defined wetland plants as plants that mainly occur in (or are exposed to) wetland habitats as described by the Ramsar Convention[17]. We summarized the 3 major groups including 42 sub-groups wetland habitat types in Ramsar Convention to be 12 categories (as estuary, intertidal wetland, mangrove swamps, rivers and lakes, brackish and saline inland wetlands, permanent non-forested wetlands, temporary non-forested wetlands, permanent forested wetlands, artificial waterbodies, marsh, bog, and fen). We collected leaf economics traits for wetland plants on a global scale including those plants exposed to intermittent/permanent wetland conditions (waterlogged or flooded) from both field and experiment measurements. The wetland plant leaf economics trait dataset was compiled based on a systematic search in Web of Science and Google Scholar (last updated on the 5th June 2018). The literature search included permutations of the following keywords: wetland plants, marsh plant, bog plant, isoetid, aquatic plants,

macrophytes, submerged plants, floating-leaved plants, emergent plants, mangroves, leaf economics traits, leaf economics spectrum, leaf nitrogen, leaf phosphorus, SLA, LMA, leaf life span, photosynthetic rate, underwater photosynthetic rate, dark respiration rate. Additionally, our network of wetland experts from around the world contributed recommendations for possible literature that we had overlooked. Finally, we added unpublished data of our own and of our network. We did not include data from other trait databases that are dominated by terrestrial records, including TRY, because the few records available for wetland plants in these databases do not have a sufficiently detailed habitat description that would allow the differentiation between waterlogged and submerged required for our analysis.

We followed the nomination system in The Plant List (http://www.theplantlist.org) to unify all plant synonyms names from the original references to a unique and consistent accepted name.

We supplemented the trait observations in our database with Ellenberg moisture indicator values. The Ellenberg moisture indicator is a classic index which generally reflects the plants' adaptation/acclimation to habitat wetness. Plant species can be categorized into 12 levels from those occupying very dry habitats (level 1) to strictly aquatic plants (level 12)[39]. For the current meta-analysis, we selected plant species with Ellenberg moisture value > 7 to represent wetland plants, as described in detail in Supplementary Methods. For these species, we selected records of the six LES traits (leaf nitrogen, leaf phosphorus, leaf dry mass per unit area, leaf life span, photosynthetic rate, and dark respiration rate). We took trait values for the same six traits for non-wetland plant traits (of 1569 species) from the GLOPNET database for comparison[3]. For a consistent analysis of trait-trait trade-offs, we expressed all leaf economics traits on a mass basis. Mass-based and area-based traits can be interconverted via a division by LMA. The mean value for each trait of each species was used (using the median did not alter the interpretation of the general pattern, as shown in Supplementary Figs. 1–3 and Supplementary Tables 3 and 4.). We used species-mean values to attain a sufficient number of trait-trait combinations for a given species. We assume that the trait observations used for calculating the species-mean values were representative for the environmental/growth conditions in which the species occurs. Possible uncertainty in species trait mean values (for example due to intra-specific variation) will then result in noise in trait-trait relationships. In total, 365 wetland species of 184 families from 151 studies were compiled and analyzed. A map of the sampling sites with accurate spatial location information can be found in Supplementary Fig. 4. The species are from varied life forms, including grasses, sedges, seagrasses, shrubs/trees, emergent, floating-leaved, isoetid, and submerged plants. Traits of most (308) species had been measured at waterlogged conditions, with submerged measurements being available for 75 species. The leaf trait data of wetland and non-wetland plant species analyzed in this paper can be found in the DRYAD repository (https://doi.org/10.5061/dryad.v6wwpzgsq).

**Statistical analysis**. The slope and its associated coefficient of determination ($R^2$) of each trait pair within the six LES traits of waterlogged and submerged wetland plants at the species level was calculated by a standardized major axis (SMA) analysis[40]. The slopes and $R^2$-values were compared to those of trait-trait relationships of non-wetland plants as derived from the GLOPNET[3]. The evaluation was based on the comparison between waterlogged wetland plants and submerged wetland plants, with non-wetland plants, respectively.

We tested each trait-trait relationship within the above-mentioned six LES traits for deviations between wetland and non-wetland plants. No test was run for the associations between leaf life span and LMA, photosynthetic rate and dark respiration rate of submerged wetland plants due to too few data points. In our SMA analysis, we conducted three tests, one to evaluate differences in slopes (i.e. steeper or shallower trait-trait relationships between wetland vs. non-wetland plants), a second to assess shift along the common slope (i.e. a more predominant position of wetland plants on either the conservative or acquisitive end of LES), and a third to assess whether trait associations of wetland and non-wetland plants can be characterized as having elevation differences among parallel slopes (suggesting a specific trait would be more -or less- costly in wetland conditions)[40]:

Test A: sma(y~x*groups) tests for differences in slopes fitted for different groups

Test B: sma(y~x+groups, shift = T) tests for a shift along the common slope

Test C: sma(y~x+groups) tests for elevation differences among parallel slopes between groups

A significant difference in slope (Test A) implies a difference in the direction and location of the relationship in trait space. Since the location and direction of lines with different slopes are not comparable[41], tests B and C were only run if there was no significantly different slope detected in Test A. If all three tests were non-significant, we conclude that wetland and non-wetland plants have similar trait-trait relationships.

The P-value is strongly depended on sample size, and it does not measure the size of an effect or the importance of a result[42]. In this study, we set a rather conservative P-value threshold ($P < 0.01$) for our tests. This was done to help reducing type I errors and to ensure that the most ecologically relevant relationships (with a reasonable effect size) were detected in these datasets[43].

The statistical analysis used R software[44]. The major axes analysis was conducted with the sma() and ma() function in the smatr package (version 3.4–8)[40].

**Reporting summary**. Further information on research design is available in the Nature Research Reporting Summary linked to this article.

## Data availability
The plant trait data that support the findings of this study and underlie its figures and tables can be downloaded from the DRYAD repository (https://doi.org/10.5061/dryad.v6wwpzgsq). The GLOPNET[3] data from which our non-terrestrial data were derived are available from: https://www.nature.com/articles/nature02403#Sec15.

## Code availability
Source code file is available in the Supplementary Software.

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

## Acknowledgements

The establishment of the wetland trait database was first discussed and started in 2008 at the Vegfunction WG39 which was funded by ARC-NZ Research Network for Vegetation Function. We would like to thank all additional contributors to this original workshop, including Paul Adam (U New South Wales, Sydney, AU), Margaret Brock (U New England, Armidale, USA), George Ganf (U Adelaide, Adelaide, AU), Irving A. Mendelssohn (Louisiana State U, Baton Rouge, USA), Eliska Rejmánkova (U California, Davis, USA), Brian Sorrell (Aarhus U, Aarhus, DK), and Evan Weiher (U Wisconsin, Eau Claire, USA). Yingji Pan is grateful to support from the China Scholarship Council (Grant No. 201606140037).

## Author contributions

P.v.B. initialized this research. Y.P., P.v.B. and E.C. designed and planned the research. Y.P. and P.v.B. compiled the data with inputs from J.A., W.A., B.R.C., T.D.C., O.P., E.J. W.V. and L.A.C.J.V. All analyses were done by Y.P. with inputs from all co-authors. Y.P., P.v.B. and E.C. wrote the first drafts of the paper, which was further improved by inputs from all co-authors.

## Competing interests

The authors declare no competing interests.
