## [Peer Review File · Nature Communications]

Reviewers' comments, first round:

Reviewer #1 (Remarks to the Author):

The authors studied leaf economics spectrum in wetland plants. They showed that the leaf economics spectrum is largely conserved among wetland species as well as among non-wetland species, but found some shift of trait-trait relationships in wetland species. The results are somewhat interesting.

Although the authors have collected many data from literatures, the number of data points are not necessarily sufficient to get reliable conclusion. For example, the authors showed that the slope of the Rmass-P relationship was positive in non-wetland plants but negative in wetland plants. We cannot know whether the latter is a general trend or caused by an outlier. Furthermore, it is not clear whether the authors were careful on the plasticity and environmental effects in leaf traits. For example, Trait values are known to vary depending on growth irradiance even within a species. Photosynthetic rate is sensitive to the environmental condition when the rate is measured. GLOPNET collected data for sun and shade leaves but only data from sun leaf were used for the analysis. GLOPNET also used photosynthetic data that were obtained at a common condition. The authors did not describe how plasticity and environmental conditions influenced obtained trait data. I suspect this is one of causes for large variation in data of wetland plants.

Minor comments

- In figures, the unit of some traits are expressed using superscripted "-1" or "-2" in some part while "/2" is used in others.

Reviewer #2 (Remarks to the Author):

(see next page)

Review for NCOMMS-20-03337-T "The leaf economics spectrum revisited: global trait patterns in wetlands"

The present study compares leaf trait covariation patterns (the leaf economics spectrum, LES) in wetland plants with those in terrestrial plants. Using global extent data compiled from the literature complemented with own data, bivariate trait-trait relationships are compared between terrestrial plants, waterlogged plants and submerged plants. The authors observe shifts in trait space and some differences in slopes of bivariate relationships. They conclude that wetland plants occupy the more acquisitive end of the LES, irrespective of costs of traits adaptive to the physicochemical peculiarities of wetlands.

The study is interesting for its exploration of the proclaimed universality of the leaf economics spectrum in its boundary regions (habitats with unique conditions). Probing the applicability of theory by means of a global dataset it will likely be of broad interest to researchers in plant functional ecology and has the potential to spark further research into wetland plant functional ecology as well as general patterns of stress-related adaptations, their costs and their effects on other functional relationships.

In its present condition, however, the paper could be improved on considerably by sharpening the argumentation and conclusions, improvement of and potential additional statistical analysis, and the so far omitted discussion of a couple of surprising findings, as suggested below.

Discuss wetland types and clarify early what "wetland plants" are being analyzed. As the authors concede themselves, wetlands are a group of habitats that differ broadly in their conditions. The Ramsar convention's definition (as referenced in the Methods section) ranges from shallow coastal waters to peatlands, which clearly differ in the type and intensity of "environmental stressors" [introduction] to which wetland plants have adapted. The introduction would benefit from a delineation of the types of wetlands analyzed here, and the range of stressors dominating within these. Similarly, the wetland plants used in the study should be introduced and described in the introduction, to enable the reader to assess results in light of this information. Presently, one only learns in the methods section at the end of the paper that the data stem from a range of life forms from seagrass to trees, and that also obligate aquatics are included (which some references might not even consider to be wetland plants). The same holds true for a definition of the groups being compared ("waterlogged" and "submerged") – these should be similarly described in the introduction to facilitate interpretation of the results.

Sharpen the language. The conclusions drawn are in places described in an imprecise and ambiguous manner that may lead to misinterpretations. In the abstract, for example, the authors write that "wetland plants show shifted trait-trait relationships" to mean a shift within trait space along the same relationship as observed in terrestrial plants. The phrase to my mind confuses shifts within trait space along a common slope (the trait-trait relationship) with shifts of the line itself (change in slope or elevation) – two very different pattern components that are furthermore both studied separately within the paper. Similarly, the conclusion presented in the abstract ("we conclude...") could be stated more concisely, and with reference to the studied LES, as e.g. "wetland plants tend to cluster at the acquisitive end of the LES". Similarly fuzzy language occurs elsewhere in the manuscript.

Interpretation and discussion of results. While SMA is an appropriate statistical approach for the stated aim, its results could be interpreted with greater care. Among the first findings presented in the results section is the observation that "overall bivariate correlations between wetland plants [...]"

were weaker than for the pairs among non-wetland plants [...], which suggests that wetland plants are less constrained within the LES with larger trait variation.” As evidence for this result, the R^2 value of bivariate relationships is shown. Two observations regarding this: First, R^2 is not independent of sample size; for smaller samples, more random variation can be expected, and smaller values do not necessarily indicate a “weaker” relationship. (An indication of the sample size for each of these comparisons would have been helpful.) Second, the R^2 values shown in Table 1 are *not* “overall weaker”: in two out of five comparisons, R^2 is lower for wetland plants than terrestrial plants, whereas in two other comparisons, the opposite is the case (R^2 is higher for wetland plants than terrestrial plants), while the remaining comparison is not significant.

Similarly, P-values are quoted to decide on the presence of significant differences in slope. P-values strongly depend on sample size, yet the authors have chosen to set the threshold for significance at the more conservative value of $P=0.01$. Relationships that would be significant at the standard threshold of $P=0.05$ in spite of smaller sample size (such as the relationship between leaf P and LMA, Fig. 1c, or LL and A_{mass} , Fig. 3d) are interpreted as non-existent. These trends could be discussed, and the choice of the lower threshold justified more explicitly.

Inexplicable to me is the omission of the discussion or even mentioning of changes in the actual *direction* of slopes (e.g. Fig. 2 e-g). These are ignored, as far as I can tell, because they again do not qualify as “significant” under a $P=0.01$ threshold, but would for a different P-value threshold. These patterns should at least be mentioned and discussed. An important treatment of P-value usage and interpretation that may be useful to the authors is Wasserstein et al. (2019).

It is furthermore unfortunate that, with a significant difference in slope, tests for shifts along the slope and changes in elevation (termed “parallel slopes”) are not conducted due to violated assumptions, as these are components of interest even with a change in slope. Perhaps there would be robust tests that would allow to also assess the latter?

Consider a multivariate analysis. The LES is inherently a multivariate pattern, concerned with the co-variation of multiple traits across species. The authors could have preceded the more detailed analysis of bivariate relationships with a multivariate exploration of all traits to situate wetland plants within n-dimensional trait space.

Figures. As a final comment, it would improve the legibility of figures if regression lines for waterlogged and submerged plants would be coloured accordingly, or their belonging to either group be indicated by other means.

Response to Reviewers

See next page

We are very grateful for your thoughtful comments and suggestions on our manuscript. Based on these comments, we have made careful modifications to the original manuscript. The individual comments and questions are responded one by one as below:

Reviewer 1:

Comment 1: Although the authors have collected many data from literatures, the number of data points are not necessarily sufficient to get reliable conclusion. For example, the authors showed that the slope of the R_{mass} -P relationship was positive in non-wetland plants but negative in wetland plants. We cannot know whether the latter is a general trend or caused by an outlier.

Response: Indeed, some trait-trait relationships are based on limited number of data points. However, with fewer data points, it is also increasingly harder to get significant relationships from a statistical point of view (as the statistical power decreases). Therefore, even with a limited number of data points, we would trust the pattern if the P value is (highly) significant ($P < 0.01$). However, if the P value is not significant, we cannot say whether this is due to the limited number of data points or caused truly by a similar trend.

Moreover, we understand that our figures might not have been clear enough and we redesigned the figures to clarify whether the lines are different and in which property they differ; we have also further clarified the figure captions (line 150, 189 & 225). The solid regression line in the figure now indicates a significant difference in the slope of the line, compared with non-wetland plants. In the case of R_{mass} -leaf P, our results suggest a similar slope for wetland and non-wetland plants, with a significant shift along the common slope in waterlogged conditions and the occurrence of two parallel slopes with a significantly different intercept in submerged conditions (line 188: Fig. 2e).

Comment 2: Furthermore, it is not clear whether the authors were careful on the plasticity and environmental effects in leaf traits. For example, Trait values are known to vary depending on growth irradiance even within a species. Photosynthetic rate is sensitive to the environmental condition when the rate is measured.

Response: We do appreciate the importance of the intra-specific variations in ecological studies. However, in this study, we had to use species-mean values in order to attain a sufficient number of trait-trait combinations for a given species (as many studies only report one trait). This implies that, while we fully acknowledge variation in trait values within species, we assume that the trait observations that have been used for calculating the species mean values were representative for the environmental/growth conditions in which the species occurs. Considering the general lack of information on environmental conditions to correct for such intra-specific variations, we could not explicitly account for intra-specific variations. The possible uncertainty in species trait mean values (due to intra-specific variation) might have been expressed in noise in trait-trait relationships. Despite this possible noise, we were able to

prove trait differences between wetland and non-wetland plants. We have indicated this in the methods section (lines 405-409).

Comment 3: GLOPNET collected data for sun and shade leaves but only data from sun leaf were used for the analysis. GLOPNET also used photosynthetic data that were obtained at a common condition. The authors did not describe how plasticity and environmental conditions influenced obtained trait data. I suspect this is one of causes for large variation in data of wetland plants.

Response: A high proportion of data from GLOPNET were indeed from sunlit leaves. While indeed in terrestrial ecosystems, the protocols for sampling leaves are highly standardized (e.g. using Cornelissen et al. 2003), this is not the case for many wetland studies (as this research community is basically decoupled from the functional ecology community developing traits-based approaches). Therefore, many studies in our wetland database do not explicitly mention whether sunlit leaves were sampled. Data from laboratory measurements commonly refer to plants grown in set-ups excluding intra- and inter-specific competition and hence excluding shading. Also many field measurements such as those by Van Bodegom et al., Ordoñez et al. and Douma et al. are known to be from sunlit leaves. We therefore assume that the effects of shading on trait values are limited in this study.

More important, the plasticity along the environmental gradient alone should only cause additional variations within species, but should not lead to bias within the inter-specific level as long as the suite of environmental conditions included in the species mean values are representative for the species. Our methodology of analyzing species-mean values thus eliminates the impacts of such environmental variations to provide robust results on trait-trait relationships. We have indicated this in the methods section (lines 405-409)

Cornelissen JHC, Lavorel S, Garnier E, Díaz S, Buchmann N, Gurvich DE, Reich PB, Ter Steege H, Morgan HD, Van Der Heijden MGA, *et al.* 2003. A handbook of protocols for standardised and easy measurement of plant functional traits worldwide. *Australian Journal of Botany* 51: 335–380.

Minor comments: In figures, the unit of some traits are expressed using superscripted "-1" or "-2" in some part while "/2" is used in others.

Response: Thank you very much for the suggestion. We have revised the figures to use superscripted "-1" or "-2" consistently.

Reviewer 2:

*Comment 1: **Discuss wetland types and clarify early what “wetland plants” are being analyzed.** As the authors concede themselves, wetlands are a group of habitats that differ broadly in their conditions. The Ramsar convention’s definition (as referenced in the Methods section) ranges from shallow coastal waters to peatlands, which clearly differ in the type and intensity of “environmental stressors” [introduction] to which wetland plants have adapted. The*

introduction would benefit from a delineation of the types of wetlands analyzed here, and the range of stressors dominating within these. Similarly, the wetland plants used in the study should be introduced and described in the introduction, to enable the reader to assess results in light of this information. Presently, one only learns in the methods section at the end of the paper that the data stem from a range of life forms from seagrass to trees, and that also obligate aquatics are included (which some references might not even consider to be wetland plants). The same holds true for a definition of the groups being compared (“waterlogged” and “submerged”) – these should be similarly described in the introduction to facilitate interpretation of the results.

Response: Thank you for the suggestion. We revised the introduction section to provide information about the wetland habitat types, life form and how we deal with the “waterlogged” and “submerged” data (line 73-86).

*Comment 2: **Sharpen the language.** The conclusions drawn are in places described in an imprecise and ambiguous manner that may lead to misinterpretations. In the abstract, for example, the authors write that “wetland plants show shifted trait-trait relationships” to mean a shift within trait space along the same relationship as observed in terrestrial plants. The phrase to my mind confuses shifts within trait space along a common slope (the trait-trait relationship) with shifts of the line itself (change in slope or elevation) – two very different pattern components that are furthermore both studied separately within the paper. Similarly, the conclusion presented in the abstract (“we conclude...”) could be stated more concisely, and with reference to the studied LES, as e.g. “wetland plants tend to cluster at the acquisitive end of the LES”. Similarly fuzzy language occurs elsewhere in the manuscript.*

Response: Thank you for the suggestion. We struggled with finding the right balance between concise phrasing and unambiguous phrasing. In the abstract, we have now solved this with rephrasing along the lines suggested by the reviewer (line 13-16). Elsewhere in the manuscript, we provide a first-time definition to the terms introduced and then stick to those terms for the remainder of the manuscript, in an attempt to solve this issue.

*Comment 3: **Interpretation and discussion of results.** While SMA is an appropriate statistical approach for the stated aim, its results could be interpreted with greater care. Among the first findings presented in the results section is the observation that “overall bivariate correlations between wetland plants [...] were weaker than for the pairs among non-wetland plants [...], which suggests that wetland plants are less constrained within the LES with larger trait variation.” As evidence for this result, the R^2 value of bivariate relationships is shown. Two observations regarding this: First, R^2 is not independent of sample size; for smaller samples, more random variation can be expected, and smaller values do not necessarily indicate a “weaker” relationship. (An indication of the sample size for each of these comparisons would have been helpful.) Second, the R^2 values shown in Table 1 are not “overall weaker”: in two out of five comparisons, R^2 is lower for wetland plants than terrestrial plants, whereas in two other comparisons, the opposite is the case (R^2 is higher for wetland plants than terrestrial plants), while the remaining comparison is not significant.*

Response: The coefficient of determination, R^2 is independent of sample size: with a change in sample size, the variance in the estimated R^2 will change, but not the expectation of the R^2 . This translates into different statistical power to detect significant relationships (i.e. different P-values), but not in different R^2 values. Therefore, we judged whether the relationships were “weaker” or “stronger” by comparing their R^2 value. We agree that an indication of sample size is helpful and we added this to Table 1. Among the significant trait-trait relationships, five out of seven relationships of waterlogged plants have a lower R^2 than that of terrestrial plants, while three out of four relationships for submerged plants have a lower R^2 than that of terrestrial plants (Table 1). That is the reason we consider that the relationships in wetland plants are overall weaker than terrestrial plants. This has now been made explicit in the results (lines 100-105).

Comment 4: Similarly, P-values are quoted to decide on the presence of significant differences in slope. P-values strongly depend on sample size, yet the authors have chosen to set the threshold for significance at the more conservative value of $P=0.01$. Relationships that would be significant at the standard threshold of $P=0.05$ in spite of smaller sample size (such as the relationship between leaf P and LMA, Fig. 1c, or LL and Amass, Fig. 3d) are interpreted as non-existent. These trends could be discussed, and the choice of the lower threshold justified more explicitly.

Response: We agree that P-value is strongly dependent on sample size, and that it does not measure the size of an effect or the importance of a result (Wasserstein et al. 2019). The statistics testing unit such as P, t and F values were developed in an era that calculations were done by hand with relatively small sample sizes. In that era, statisticians arbitrarily set the $P<0.05$ as the threshold for significance based on the sample size commonly used at that time, to coincide with a reasonable effect size. With large sample sizes, which are much more common these days, almost any relationship will be statistically significant (Nakagawa & Cuthill 2007). The below graph shows the relationships between R^2 , sample size at two different P thresholds – e.g. a dataset with 77 samples and a relationship of $R^2 = 0.05$ would be significant at $P = 0.05$, whereas at $P = 0.01$ – a dataset with the same relation would need to contain 131 samples to be considered significant. As more and more research contains a much larger sample size, we are in need of a more conservative threshold for the P value that concur with a reasonable and ecologically relevant effect size.

In our research, we compiled large datasets for trait-trait relationships such as leaf N-LMA, leaf P-LMA, leaf P-leaf N. Considering the relatively large sample size involved, we prefer a more conservative P value (as $P < 0.01$) to reduce a type I error. This choice, while we consider it valid for most of our comparisons, may have led to conservative estimates of significance in cases with only a limited number of data points, such as leaf life span- R_{mass} and leaf life span- A_{mass} . In those situations, we may have been more conservative than perhaps necessary, by using our default choice of $P < 0.01$. However, we preferred that over using two standards: $P < 0.05$ for smaller datasets and $P < 0.01$ for larger datasets as that introduces some arbitrariness. Therefore, we finally chose one standard conservative threshold at $P < 0.01$. This justification has been added to the manuscript (line 345–353 & line 441–445).

Nakagawa S, Cuthill IC. 2007. Effect size, confidence interval and statistical significance: A practical guide for biologists. *Biological Reviews* 82: 591–605.

Wasserstein RL, Schirm AL, Lazar NA. 2019. Moving to a World Beyond “ $p < 0.05$ ”. *American Statistician* 73: 1–19.

Comment 5: Inexplicable to me is the omission of the discussion or even mentioning of changes in the actual direction of slopes (e.g. Fig. 2 e-g). These are ignored, as far as I can tell, because they again do not qualify as “significant” under a $P=0.01$ threshold, but would for a different P-value threshold. These patterns should at least be mentioned and discussed. An important treatment of P-value usage and interpretation that may be useful to the authors is Wasserstein et al. (2019).

Response: See also response to Comment 4 above related to our choice of p-value. Moreover, we have probably not been explicit enough in explaining the meaning of the solid regression line in Fig. 2. In our original submission, a solid line indicates a significant difference either in slope, shift or the occurrence of parallel slopes (or a combination thereof). For example, in the case of R_{mass} -leaf P (Fig. 2e), it only suggests a significant shift along the common slope in

waterlogged and the occurrence of two parallel slopes with a significantly different intercept in submerged conditions. However, the slopes are similar ($P > 0.05$). Moreover, all the slopes analyzed in this study showed the same direction (as in positive or negative) with non-wetland plants (line 108 & 118: Table 1 & 2). The remark of the reviewer shows that this use of solid lines has been confusing. In an attempt to clarify this, we have redesigned the lines and colors of Fig. 1-3 and added more explicit captions on the differences to avoid further misunderstanding (line 150, 189 & 225).

Comment 6: It is furthermore unfortunate that, with a significant difference in slope, tests for shifts along the slope and changes in elevation (termed “parallel slopes”) are not conducted due to violated assumptions, as these are components of interest even with a change in slope. Perhaps there would be robust tests that would allow to also assess the latter?

Response: We use the standardized major axis (SMA) method to analyze the data. According to the developers of the method, they recommend that if there is a significant difference in slope, one should not test the shift along the slope nor changes in elevations (termed ‘parallel slopes’ in our paper). See: Warton et al. 2006 : “It should be noted that if the slope of the relationship does change across sites, then Questions c (“is there a shift in elevation across these sites?”) and d (“is there a shift along a common axis for different sites?”) cannot be addressed. This is for the same reasons as in analysis of covariance – elevation and location along the line are not comparable for lines with different slopes”. So, while a significant difference in slope implies some differences in elevation and a shift along the slope, unfortunately, the above implies there is no robust test available that can deal with such Type II regressions. This has been made more explicit in the text (line 441-445 in Methods)

Warton DI, Wright IJ, Falster DS, Westoby M. 2006. Bivariate line-fitting methods for allometry. *Biological Reviews of the Cambridge Philosophical Society* 81: 259–291.

Comment 7: Consider a multivariate analysis. The LES is inherently a multivariate pattern, concerned with the covariation of multiple traits across species. The authors could have preceded the more detailed analysis of bivariate relationships with a multivariate exploration of all traits to situate wetland plants within n-dimensional trait space.

Response: Thank you for the suggestion. We are taking the species-mean of measurements from different papers and data sources to allow as many bivariate trait-trait relationships as possible in the analysis. However, even in this way, we cannot gain sufficient data to put all the six traits in one multivariate analysis (i.e. many species in our database only have trait observations for a small set of traits). Therefore analysis such as NMDS and non-parametric MANOVA are unfortunately not possible in our case.

Comment 8: Figures. As a final comment, it would improve the legibility of figures if regression lines for waterlogged and submerged plants would be colored accordingly, or their belonging to either group be indicated by other means.

Response: Thank you for the suggestion. We have revised the figures to be in color (line 150, 189 & 225). This also allowed us to better distinguish in the figure the differences in shifts along a common slope vs. parallel slopes vs. differences in slopes.

Reviewer Comments, second round

Reviewer #2 (Remarks to the Author):

Thank you for considering and addressing my previous comments. The resubmission of this manuscript shows a better developed text and particularly figures and figure captions have been improved considerably.

Most of my earlier comments have been addressed adequately, except regarding the statistical analysis and the interpretation of the results (labelled comment 3 in the rebuttal letter). I maintain that the problem of sample size remains, as elaborated in the following.

The authors use R^2 to assess the strength of the relationship between bivariate trait relationships. That in itself is appropriate. My previous comment read: "Among the first findings presented in the results section is the observation that "overall bivariate correlations between wetland plants [...] were weaker than for the pairs among non-wetland plants [...], which suggests that wetland plants are less constrained within the LES with larger trait variation." As evidence for this result, the R^2 value of bivariate relationships is shown. Two observations regarding this: First, R^2 is not independent of sample size; for smaller samples, more random variation can be expected, and smaller values do not necessarily indicate a "weaker" relationship. (An indication of the sample size for each of these comparisons would have been helpful.)"

I would like to thank the authors for adding sample sizes to Table 1. These values underscore, however, my concern. The authors' response was as follows:

"Response: The coefficient of determination, R^2 is independent of sample size: with a change in sample size, the variance in the estimated R^2 will change, but not the expectation of the R^2 . This translates into different statistical power to detect significant relationships (i.e. different P-values), but not in different R^2 values. Therefore, we judged whether the relationships were "weaker" or "stronger" by comparing their R^2 value."

This is in itself correct: the expectation of the R^2 will not change with sample size – however, the expectation of R^2 is not quantifiable by any other means than the sample at hand, it can only be estimated. In other words, the quantity reported is not the expectation of R^2 , but by necessity the estimated R^2 , which does depend on sample size (precisely because of its increased variability at smaller sample sizes that is mentioned). In the comparison of terrestrial plants with wetland plants, the datasets available for terrestrial plants always have considerably larger sample sizes (now shown in Table 1, e.g. $n=1322$ for terrestrial plants compared to $n=42$ for submerged plants and $n=178$ for waterlogged plants in the first comparison, and these n are among the highest for wetland plants). I am concerned with the robustness of the interpretation of R^2 as strength of the relationship, and the comparison thereof between groups, when group sizes differ that much.

Similarly, the absence of significant differences in bivariate slopes between groups (SMA Test A) is interpreted as a common slope. The threshold to decide on statistical significance is chosen to be $p < 0.01$. Technically, this is correct. But in terms of ecological interpretation, the influence of sample size on this assessment should be considered carefully, ideally discussed. For example, in wetland plants, the relationships of dark respiration rate R_{mass} with several other traits show slope estimates that differ in sign (direction) compared to terrestrial plants (as was also commented on by reviewer #1). None of these slope estimates differ significantly from terrestrial plants (Table 1), but that is hardly surprising considering the sample sizes of $n=10$ and $n=16$ for the wetland plant groups and the conservative p-Value. Whether the absence of a significant slope (at $p < 0.01$) in these small samples is sufficient to assume a common slope is a matter of interpretation.

Perhaps it could be discussed more openly and transparently that the data is insufficient to reject the null hypothesis of common slopes, but that it equally well may be insufficient to support the conclusion of common slopes. (And please refrain from calling the dataset "large", as is done in several instances, when this is only the case for some but not all of the traits assessed here – the smallest group size is a questionable $n=3$.)

The culture of overstating confidence in results in order to publish in high impact journals (see

Vinkers et al. 2015) is unfortunate. I think this study is interesting and exciting and well-worth publishing, even in light of the uncertainties associated with data limitations. Just please address them openly.

Vinkers, C. H., Tjeldink, J. K., & Otte, W. M. (2015). Use of positive and negative words in scientific PubMed abstracts between 1974 and 2014: Retrospective analysis. *BMJ (Online)*, 351(December), 1–6. <https://doi.org/10.1136/bmj.h6467>

Editor's note: Reviewer 3 disagrees that certain sample sizes are too small here, but does recommend that the issues raised are discussed in a few well-chosen sentences. The referee also asks to carefully review the slope data in Table 1 (which confidence intervals overlap and which ones do not?) and, importantly, asks that you do not consider just the slope value. The reviewer suggests considering to set a clear rule for running slope comparisons, say $p < 0.15$, even though this is inevitably subjective. If SMA slopes are fitted and compared to each other even when correlation approaches zero, it should be acknowledged that the tests are provision given lack of confidence on the nature of those slopes (including the sign).

Reviewer #3 (Remarks to the Author):

I have come on board to review the revised manuscript (and assess the responses to previous reviews), not having seen the manuscript before.

I am generally satisfied that the authors attended effectively to the previous reviews.

This study has clear potential to make a useful contribution to plant ecology. LES relationships are poorly understood for wetland plants, and the authors have made a serious effort to draw together all available data to facilitate a reliable quantification of key trait-trait relationships, and to assess the extent to which they differ from relationships previously reported for non-wetland plants. The short answer is that relationships are generally similar, but there are some interesting slope offsets (in elevation) and a general tendency for wetland species to fall further towards the "fast-return" (or "acquisitive") end of key LES relationships. Some of the reported relationships are based on rather few data points (as clearly detailed by the authors), but (firstly) that's unavoidable, and (secondly) the importance of these sorts of studies is that they set a context for all future work, and suggest fruitful research agenda. For example, here the authors report generally higher P in waterlogged species, and lower dark respiration rate at a given photosynthetic rate, or leaf N, or LMA. Prospective explanations are offered here, but future work could serve to nail down the mechanisms responsible (or, alternatively, quantify whether these patterns are robust in the face of more data).

My only substantial comment concerns multiple misuses of the term "parallel" in regards to SMA slope tests:

L434. "Test C: $\text{sma}(y \sim x + \text{groups})$ tests for parallel slopes between groups".

Here, and right throughout Results, describing this test and the related phenomenon as "parallel slopes" is unhelpful and actually also incorrect. If test A shows that the groups do not differ in their SMA slopes, then the three slopes can already be described as effectively "parallel". Test C would be better described as a test "for elevation (or intercept) differences among parallel slopes". It is the offset between similar (parallel) slopes that is being tested here, not the "parallelness" itself.

A few other instances where this causes confusion include (but are not limited to):

L118-119 "... and a change in elevation resulting in parallel slopes (Par.)" [this does not make sense. The slopes are parallel, whether or not there is an elevation difference].

L123-124 "...the occurrence of parallel slopes cannot be tested". [term is misused here]

L154-155 "*par. Identif[ies] ... a significantly different intercept resulting in a parallel slope, respectively.". [this is incorrect, because it is not the different intercept which "results" in a parallel slope. The slopes were already deemed as effectively parallel, by passing test A]

** I note that at L130-132 (and a couple of other places) the authors do indeed describe the phenomenon and test correctly: "However, the parallel slopes of both waterlogged and submerged wetland plants were elevated compared to non-wetland plants (both $P < 0.001$), which indicates that at a given leaf N, wetland plants tended to have a higher leaf P than non-wetland plants" **

Some minor comments:

L29. Aarea should not be included in this list of traits that vary globally with LMA. In the glopnet dataset Aarea and LMA were unrelated ($r^2 = 0.003$; $P = 0.153$). Perhaps change text to "...and lower photosynthetic rates, at least on a mass basis (A_{mass})".

L240-249. This summary of main results is useful, but I'd suggest more prominence should be given to the various types of slope offsets affecting pretty much all relationships involving Rd, in Figure 2. This seems like an especially interesting finding.

L438. Change "insignificant" to "non-significant". (Insignificant means unimportant' non-significant means "not statistically significant", which is a subtly different concept).

Ian Wright, June 2020.

REVIEWERS' COMMENTS:

Reviewer #2 (Remarks to the Author):

Thank you for considering and addressing my previous comments. The resubmission of this manuscript shows a better developed text and particularly figures and figure captions have been improved considerably.

Response: Thank you for your compliments.

Most of my earlier comments have been addressed adequately, except regarding the statistical analysis and the interpretation of the results (labelled comment 3 in the rebuttal letter). I maintain that the problem of sample size remains, as elaborated in the following.

The authors use R^2 to assess the strength of the relationship between bivariate trait relationships. That in itself is appropriate. My previous comment read: "Among the first findings presented in the results section is the observation that "overall bivariate correlations between wetland plants [...] were weaker than for the pairs among non-wetland plants [...], which suggests that wetland plants are less constrained within the LES with larger trait variation." As evidence for this result, the R^2 value of bivariate relationships is shown. Two observations regarding this: First, R^2 is not independent of sample size; for smaller samples, more random variation can be expected, and smaller values do not necessarily indicate a "weaker" relationship. (An indication of the sample size for each of these comparisons would have been helpful.)"

I would like to thank the authors for adding sample sizes to Table 1. These values underscore, however, my concern. The authors' response was as follows:

"Response: The coefficient of determination, R^2 is independent of sample size: with a change in sample size, the variance in the estimated R^2 will change, but not the expectation of the R^2 . This translates into different statistical power to detect significant relationships (i.e. different P-values), but not in different R^2 values. Therefore, we judged whether the relationships were "weaker" or "stronger" by comparing their R^2 value."

This is in itself correct: the expectation of the R^2 will not change with sample size – however, the expectation of R^2 is not quantifiable by any other means than the sample at hand, it can only be estimated. In other words, the quantity reported is not the expectation of R^2 , but by necessity the estimated R^2 , which does depend on sample size (precisely because of its increased variability at smaller sample sizes that is mentioned). In the comparison of terrestrial plants with wetland plants, the datasets available for terrestrial plants always have considerably larger sample sizes (now shown in Table 1, e.g. $n=1322$ for terrestrial plants compared to $n=42$ for submerged plants and $n=178$ for waterlogged plants in the first comparison, and these n are among the highest for wetland plants). I am concerned with the robustness of the interpretation of R^2 as strength of the relationship, and the comparison thereof between groups, when group sizes differ that much.

Response: Thank you very much for the suggestions. We calculated the confidence interval of R^2 at the 95% significance level to determine the variation of R^2 as affected by the sample size to evaluate the robustness of the interpretation of R^2 and its differences. Among the significant trait-trait relationships with no overlaps in the confidence interval of R^2 , four out of five waterlogged trait comparisons and two out of two submerged trait comparisons have lower R^2 than the corresponding

trait-trait relationship in non-wetland plants. In addition, four non-significant slopes (in itself a possible indication of a weak relationship, depending on statistical power) of waterlogged trait comparisons and four non-significant slopes of submerged conditions also showed no overlap with the R^2 observed for the corresponding trait-trait relationship of non-wetland plants. We believe this supports our conclusion that trait-trait relationships are weaker in wetland plants than non-wetland plants. The data are provided in the Supplementary Information Table 1.

Similarly, the absence of significant differences in bivariate slopes between groups (SMA Test A) is interpreted as a common slope. The threshold to decide on statistical significance is chosen to be $p < 0.01$. Technically, this is correct. But in terms of ecological interpretation, the influence of sample size on this assessment should be considered carefully, ideally discussed. For example, in wetland plants, the relationships of dark respiration rate R_{mass} with several other traits show slope estimates that differ in sign (direction) compared to terrestrial plants (as was also commented on by reviewer #1). None of these slope estimates differ significantly from terrestrial plants (Table 1), but that is hardly surprising considering the sample sizes of $n = 10$ and $n = 16$ for the wetland plant groups and the conservative p-Value. Whether the absence of a significant slope (at $p < 0.01$) in these small samples is sufficient to assume a common slope is a matter of interpretation.

Response: Thank you very much for the suggestions. We agree that part of the non-significant slopes might be due to a lack of statistical power, which we neglected previously. We therefore calculated the differences between the slopes among traits of wetland plants vs. the slopes among traits of non-wetland plants. Indeed, in particular occasions, slope differences were considerable (i.e. bigger than 1 or smaller than -1) while still being insignificant. Each of those occasions related to R_{mass} . We agree that there is indeed an issue with lack of statistical power in the trait of dark respiration rate. We revised the manuscript accordingly. Moreover, we added a new table to supplementary material showing the differences in slopes according to the SMA. We also rewrote the results section on R_{mass} to incorporate these new interpretations.

Perhaps it could be discussed more openly and transparently that the data is insufficient to reject the null hypothesis of common slopes, but that it equally well may be insufficient to support the conclusion of common slopes. (And please refrain from calling the dataset “large”, as is done in several instances, when this is only the case for some but not all of the traits assessed here – the smallest group size is a questionable $n = 3$.)

Response: Following the previous comment, we have now indicated in the result section that analyses with dark respiration rate and leaf life span are with limited datasets, which may be insufficient to reject the null hypothesis of common slopes.

The culture of overstating confidence in results in order to publish in high impact journals (see Vinkers et al. 2015) is unfortunate. I think this study is interesting and exciting and well-worth publishing, even in light of the uncertainties associated with data limitations. Just please address them openly.

Vinkers, C. H., Tjink, J. K., & Otte, W. M. (2015). Use of positive and negative words in scientific PubMed abstracts between 1974 and 2014: Retrospective analysis. *BMJ (Online)*, 351(December), 1–6. <https://doi.org/10.1136/bmj.h6467>

Response: Thank you for your suggestions. We have revised the manuscript accordingly.

****Editor's note:** Reviewer 3 disagrees that certain sample sizes are too small here, but does recommend that the issues raised are discussed in a few well-chosen sentences. The referee also asks to carefully review the slope data in Table 1 (which confidence intervals overlap and which ones do not?) and, importantly, asks that you do not consider just the slope value. The reviewer suggests considering to set a clear rule for running slope comparisons, say $p < 0.15$, even though this is inevitably subjective. If SMA slopes are fitted and compared to each other even when correlation approaches zero, it should be acknowledged that the tests are provision given lack of confidence on the nature of those slopes (including the sign).**

Response: In response to this suggestion and Reviewer 2's comments, we have carefully reviewed the slope data as indicated in our response to Reviewer 2's comments. We preferred to have an analysis with clear evaluation of the confidence in our statements, while accounting for sample size, instead of setting a priori rule for slope comparison, as we considered this to be more robust and objective.

Reviewer #3 (Remarks to the Author):

I have come on board to review the revised manuscript (and assess the responses to previous reviews), not having seen the manuscript before.

I am generally satisfied that the authors attended effectively to the previous reviews.

This study has clear potential to make a useful contribution to plant ecology. LES relationships are poorly understood for wetland plants, and the authors have made a serious effort to draw together all available data to facilitate a reliable quantification of key trait-trait relationships, and to assess the extent to which they differ from relationships previously report for non-wetland plants. The short answer is that relationships are generally similar, but there are some interesting slope offsets (in elevation) and a general tendency for wetland species to fall further towards the "fast-return" (or "acquisitive") end of key LES relationships. Some of the reported relationships are based on rather few data points (as clearly detailed by the authors), but (firstly) that's unavoidable, and (secondly) the importance of these sorts of studies is that they set a context for all future work, and suggest fruitful research agenda. For example, here the authors report generally higher P in waterlogged species, and lower dark respiration rate at a given photosynthetic rate, or leaf N, or LMA. Prospective explanations are offered here, but future work could serve to nail down the mechanisms responsible (or, alternatively, quantify whether these patterns are robust in the face of more data).

Response: Thank you for the appreciation of our analysis. We fully agree that for some relations future work is necessary to evaluate their robustness and underlying mechanisms. We added a remark along such lines to our discussion section.

My only substantial comment concerns multiple misuses of the term "parallel" in regards to SMA slope tests:

L434. "Test C: $\text{sma}(y \sim x + \text{groups})$ tests for parallel slopes between groups".

Here, and right throughout Results, describing this test and the related phenomenon as “parallel slopes” is unhelpful and actually also incorrect. If test A shows that the groups do not differ in their SMA slopes, then the three slopes can already be described as effectively “parallel”. Test C would be better described as a test “for elevation (or intercept) differences among parallel slopes”. It is the offset between similar (parallel) slopes that is being tested here, not the “parallelness” itself.

Response: Thank you very much for the suggestion. We have revised the manuscript accordingly (both at the instances outlined below and elsewhere).

A few other instances where this causes confusion include (but are not limited to):

L118-119 “... and a change in elevation resulting in parallel slopes (Par.)” [this does not make sense. The slopes are parallel, whether or not there is an elevation difference].

Response: Revised.

L123-124 “...the occurrence of parallel slopes cannot be tested”. [term is misused here]

Response: Revised.

L154-155 “*par. Identif[ies] ... a significantly different intercept resulting in a parallel slope, respectively.”. [this is incorrect, because it is not the different intercept which *results” in a parallel slope. The slopes were already deemed as effectively parallel, by passing test A] ** I note that at L130-132 (and a couple of other places) the authors do indeed describe the phenomenon and test correctly: “However, the parallel slopes of both waterlogged and submerged wetland plants were elevated compared to non-wetland plants (both $P < 0.001$), which indicates that at a given leaf N, wetland plants tended to have a higher leaf P than non-wetland plants” **

Response: Revised.

Some minor comments:

L29. Aarea should not be included in this list of traits that vary globally with LMA. In the glopnet dataset Aarea and LMA were unrelated ($r^2 = 0.003$; $P = 0.153$). Perhaps change text to “...and lower photosynthetic rates, at least on a mass basis (A_{mass})”.

Response: Revised according to the suggestion of the reviewer.

L240-249. This summary of main results is useful, but I’d suggest more prominence should be given to the various types of slope offsets affecting pretty much all relationships involving R_d , in Figure 2. This seems like an especially interesting finding.

Response: we considered this a highly interesting finding indeed and we suggest a number of possible mechanisms. On the other hand, as discussed above, our statistical power for evaluating R_d -patterns is rather limited and was for those reasons criticized by Reviewer 2. We therefore considered not to over-emphasize this finding, although the finding for the relationship between R_d and leaf N and P are robust despite the low sample size. We also highlight the need for future work to further evaluate these relationships.

L438. Change “insignificant” to “non-significant”. (Insignificant means unimportant’ non-significant means “not statistically significant”, which is a subtly different concept).

Response: Revised.

Ian Wright, June 2020.